# High-resolution mapping of cancer cell networks using co-functional interactions

Evan A Boyle[1] , Jonathan K Pritchard[1,2,3] & William J Greenleaf[1,4,*]

## Abstract

Powerful new technologies for perturbing genetic elements have recently expanded the study of genetic interactions in model systems ranging from yeast to human cell lines. However, technical artifacts can confound signal across genetic screens and limit the immense potential of parallel screening approaches. To address this problem, we devised a novel PCA-based method for correcting genome-wide screening data, bolstering the sensitivity and specificity of detection for genetic interactions. Applying this strategy to a set of 436 whole genome CRISPR screens, we report more than 1.5 million pairs of correlated "co-functional" genes that provide finer-scale information about cell compartments, biological pathways, and protein complexes than traditional gene sets. Lastly, we employed a gene community detection approach to implicate core genes for cancer growth and compress signal from functionally related genes in the same community into a single score. This work establishes new algorithms for probing cancer cell networks and motivates the acquisition of further CRISPR screen data across diverse genotypes and cell types to further resolve complex cellular processes.

**Keywords** CRISPR; functional genomics; genetic interactions; genome-wide perturbation; network topology
**Subject Categories** Chromatin, Epigenetics, Genomics & Functional Genomics; Computational Biology; Network Biology
**Mol Syst Biol. (2018) 14: e8594**

## Introduction

Understanding the complex biological underpinnings of human disease has long been a goal of network biologists (Barabási & Oltvai, 2004; Barabási *et al*, 2011). Because genes vary in their role and importance across diverse cell types, it has become increasingly clear that characterizing tissue- and cell type-specific regulation of chromatin accessibility (Roadmap Epigenomics Consortium *et al*, 2015; Breeze *et al*, 2016), chromosome looping (Javierre *et al*, 2016;

Mumbach *et al*, 2017), and gene expression (GTEx Consortium *et al*, 2017) will be central to developing a coherent understanding of disease etiology. Differences in biological pathway importance across tissues are especially vexing when modeling diseases such as cancer that specifically exploit tissue-specific pathways and preferentially acquire mutations to regulate them. Set against these challenges, the advent of new genetic perturbation systems scalable to the size of the human genome offers an unprecedented opportunity for the study of cancer cell networks and associated tissue-specific signaling paradigms that do not exist in single-celled model organisms like yeast (Gilmore, 2006; Fontana *et al*, 2010).

Genome-wide CRISPR screens (Shalem *et al*, 2014; Wang *et al*, 2014) have already enabled insights into cell trafficking (Gilbert *et al*, 2014), drug mechanism of action (Shalem *et al*, 2014; Wang *et al*, 2014; Doench *et al*, 2016), and infectious disease (Park *et al*, 2017; Gavory *et al*, 2018). Yet, while these methods allow every gene to be perturbed and scored for its effect on a phenotype of interest, this score does not provide direct insight into the logic of the biological pathways involved in mediating the phenotype. Instead, these screens report one-dimensional vectors of values: Each gene falls on a single spectrum from dis-enriched to enriched. Determining the cellular logic that integrates effects across genes requires either specialized experimental design or extensive post-processing of high-throughput screen data. Presently, there are three prominent strategies for functionally characterizing genes genome-wide based on observations from high-throughput screens: First, hit genes are frequently enriched in curated gene sets that reflect known biological processes or cellular components; second, combinations of genetic perturbations can yield non-additive effects that describe how information flows through sets of genes; and lastly, pairs of genes that exhibit correlated effects across diverse cell lines or conditions often lie in the same cell pathways. Enrichment in curated gene sets can clarify the biological processes involved by implicating cell pathways or compartments, but interpreting enrichments can be extremely difficult (Rhee *et al*, 2008; Timmons *et al*, 2015) and new experimental methods have facilitated the pursuit of the other strategies.

Combinatorial sgRNA platforms (Bassik *et al*, 2013; Kampmann *et al*, 2015) disrupt multiple genes per cell to identify epistatic interactions that can unmask gene logic. These platforms are the

---

1 Department of Genetics, Stanford University, Stanford, CA, USA
2 Department of Biology, Stanford University, Stanford, CA, USA
3 Howard Hughes Medical Institute, Stanford, CA, USA
4 Chan Zuckerberg Biohub, San Francisco, CA, USA
*Corresponding author. Tel: +1 650 725 3672; E-mail: wjg@stanford.edu

successors of double-knockout array technology used in yeast to identify genetic interactions for 90% of all genes (Tong *et al*, 2004; Costanzo *et al*, 2010, 2016). While these methods have the power to directly test hypothesized genetic interactions, technological constraints have limited individual combinatorial sgRNA studies to measuring interactions for only a small fraction of all pairs of human genes (Ogasawara *et al*, 2015; Wong *et al*, 2016; Han *et al*, 2017; Shen *et al*, 2017; Horlbeck *et al*, 2018). Ascertaining appropriate gene pairs for such phenotyping is not trivial, especially for synthetic interactions where neither genetic perturbation exhibits an effect on its own, although algorithms to overcome this challenge are under development (Medina & Goodin, 2008; preprint: Deshpande *et al*, 2017). Interpretation of measured interactions is also complicated by the fact that the extent to which genetic interactions persist across cell types or samples is unknown.

Parallel screening designs approach the identification of interacting genes in a fundamentally different manner. All genes of interest, potentially comprising the entire genome, are screened in a diverse panel of cell lines, and the perturbation effect sizes across these cell lines are recorded as a gene perturbation profile for every gene (Fig 1A and B). The distinct genetic and epigenetic features of each cell line modify its susceptibility to disruption of pathways, organelles, or even individual protein complexes. In general, two genes that have correlated gene perturbation scores across many cell lines are inferred to be functionally related, with greater correlation implying greater shared function, an observation first made in yeast (Fraser & Plotkin, 2007). More recently, data from only 6 cell lines sufficed to verify essential gene pathways (Hart *et al*, 2015); however, a larger panel of 14 parallel AML line CRISPR screens allowed more systematic validation of cancer metabolic and signaling pathways (Wang *et al*, 2017). The publication of hundreds of CRISPR screens in cell lines drawn from diverse cell lineage and mutational backgrounds has invited even broader surveys of co-essentiality (Meyers *et al*, 2017; Data ref: Meyers *et al*, 2017). One study has dissected the composition of essential protein complexes (Pan *et al*, 2018), another has leveraged the natural occurrence of gene activating mutations to ascertain likely genetic interactions (Rauscher *et al*, 2018), and other work accessible as a preprint has focused on the organization of cancer growth pathways (preprint: Kim *et al*, 2018). In all these cases, interactions identified from correlated gene profiles operated on multiple levels of cellular regulation, validating parallel screening as a powerful tool for reconstructing cell networks.

While effective, parallel screening approaches require more substantial post-processing of results than combinatorial screens. Studies involving parallel screens are straightforward to design, but technical variation in how the screens are performed as well as copy number variation across cell backgrounds can confound the results (Zhang & Lu, 2009; Aguirre *et al*, 2016). Recent work has shown that copy number variation can underlie the strongest hits in CRISPR-knockout screens, and multiple groups have proposed corrective algorithms to confront this problem (Pommier, 2006; Meyers *et al*, 2017; Data ref: Meyers *et al*, 2017; preprint: Wu *et al*, 2018). Additional heuristics aimed at increasing the quality of genetic interactions identified from parallel genetic screens have included discarding entire screens with noisy effect sizes, setting an effect size threshold for correlating genes, and capping the number of interactions per gene (Wang *et al*, 2017; preprint: Kim *et al*,

2018; Pan *et al*, 2018); however, reliance on these heuristics prevents truly unbiased genome-wide analyses (McFarland *et al*, 2018). Furthermore, as the scale and diversity of published genetic screens grow, so will the need for new statistical techniques that can correct for technical variation while preserving even small levels of true signal.

In this work, we develop a flexible, unsupervised approach for removing confounding from parallel genome-wide CRISPR screens. We apply this approach to the 436 CRISPR screens of Project Achilles and compute corrected gene profiles for all reported genes. We identify more than 1.5 million pairs of significantly correlated co-functional genes, substantially more than reported by other studies. Finally, we detect functionally delineated gene communities and characterize their specificity with respect to cell lineage and mutational backgrounds. Using these gene communities, we provide new insight into cancer cell network topology including scores for each gene's potential to drive a pathway important for cancer proliferation.

## Results

### Correcting for technical confounding found in parallel genetic screen data

We first downloaded CRISPR screen gene summary data corrected for copy number confounding from the Project Achilles data depository (Meyers *et al*, 2017; Data ref: Meyers *et al*, 2017) and matched RNA-Seq and mutation data from the Cancer Cell Line Encyclopedia (CCLE) website (Barretina *et al*, 2012). The results of the CRISPR screens form a matrix, where each row in the matrix serves as a gene essentiality profile that summarizes the knockout phenotype of the gene across the 436 cell lines (the columns of the matrix). As reported by others, the degree to which two gene essentiality profiles (rows) are correlated reflects their functional relationship. This functional relationship can reflect many gene–gene relationships, including membership in the same metabolic pathway or protein complex, and depends on the mutations present in the cell lines tested (Fig 1A and B; Wang *et al*, 2017; Pan *et al*, 2018).

Because technical factors in the dataset could drastically skew Pearson's correlation values, we explored a large set of control genes expected to have little to no phenotype in the context of cancer proliferation: olfactory receptors [as classified by the HUGO Gene Nomenclature Committee (HGNC)]. Some olfactory receptors shared identical sgRNAs, and in these cases, we retained only one olfactory receptor to avoid duplicate gene profiles. Within this set, all pairwise correlations were calculated under the expectation that strong effect sizes would indicate technical confounding. Under a model of uniformly null phenotypes, we would expect correlations to be tightly distributed around 0. In fact, olfactory receptors often exhibited profiles that were highly correlated across genetic backgrounds, strongly suggesting the presence of technical confounding.

To investigate the unanticipated signal of essentiality in olfactory receptors, we evaluated the covariance of the knockout growth phenotypes across cell lines using principal components analysis (PCA). We performed PCA on the matrix of growth defects from each olfactory receptor (row) and cell line background (column;

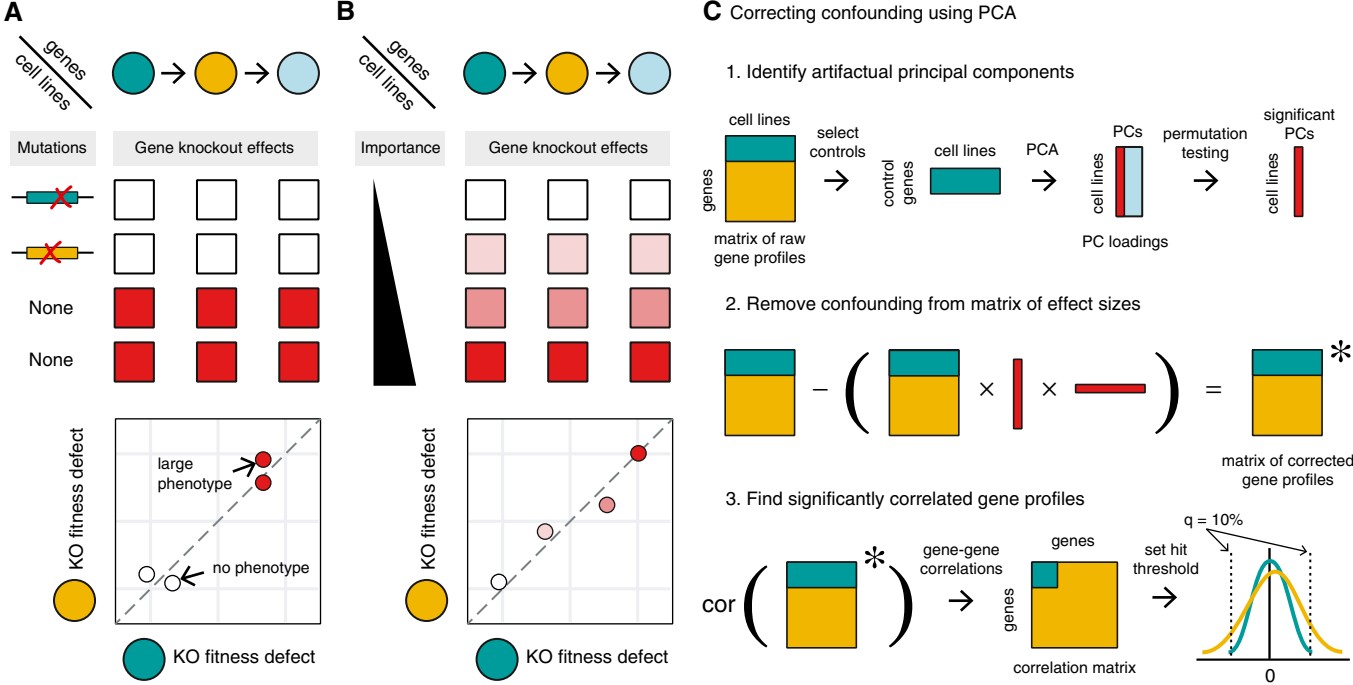

**Figure 1. Detecting co-functional interactions from parallel genetic perturbation screens.**

A, B   Diversity in (A) mutational background and (B) pathway activity produce correlated gene essentiality profiles of knockout effect sizes for members of the same pathway. Cell lines with inactivating mutations in or downregulation of biological pathways are impervious to gene knockout. Discrete and continuous differences across cell lines can thus be summarized by a correlation coefficient.

C   Nonspecific sources of variation across cell lines can be removed by learning principal components from a set of control genes that should not contain biological signal and subtracting these components from the raw gene profiles.

Fig 1C). The resulting principal components describe how likely each cell line is to exhibit essentiality among olfactory receptors. If the essentiality measurement of each olfactory receptor was being driven by gene function or chromosomal locus, these profiles—a single number per cell line—would be expected to be poorly predictive genome-wide. However, we found that the top five principal components explained significantly more variance than expected by permutation testing, with over half of the variance explained by the first principal component alone (Fig EV1A). We repeated this approach with a curated set of nonessential genes in place of olfactory receptors and reproduced the loadings on the first principal component, demonstrating that signatures are robust to choice of control gene set ($R = 0.92$, Fig EV1B).

The loadings on the first principal component, which again explained most of the variance in olfactory receptor scores, were highly correlated with the variance in effect size estimates for olfactory receptor genes for each cell line ($R = -0.89$, Fig EV1C). One possibility is that regressing on the variance of each cell line acts as a technical correction, re-centering and scaling the effect sizes in a manner similar to that performed in a similar study (Rauscher *et al*, 2018) and recommended in a recent article (McFarland *et al*, 2018). In this case, our scaling would roughly equalize variance in biological effect sizes among negative control genes. Alternatively, something that would be well captured by a single number per cell line such as cell health or sensitivity to double-strand breaks would also be consistent with prior work investigating CRISPR screen

specificity and could produce the same relationship with variance in cell line effect sizes (Morgens *et al*, 2017; Rosenbluh *et al*, 2017). In this case, one might expect mutational status in certain genes to predict the loading on cell line signatures, but the presence of coding or loss-of-function mutations in all recorded genes was not associated with principal component loadings (Wilcoxon rank-sum test, $q < 0.1$ for all genes). In any case, re-centering and scaling is not likely to underlie the four other orthogonal candidate signatures of confounding.

To produce an improved dataset, we first subtracted all five candidate signatures of confounding from each gene's essentiality profile (Fig 1C). Using these corrected gene essentiality profiles, we again computed the correlation of all pairs of genes. In many cases, as seen for peroxisome genes, learned relationships before and after correction appeared unchanged, but in others, as seen for spliceosome scaffold proteins, previously unremarkable sets of genes appeared tightly related (Fig 2A and B). Pairs of olfactory receptors that were persistently correlated after correction were often in very close physical proximity on chromosome arms. As observed previously (Meyers *et al*, 2017; Data ref: Meyers *et al*, 2017), physical proximity often increases the rate of correlation even after CERES copy number correction (Fig EV1D), suggesting either that there are differences in local toxicity to dsDNA breaks or that copy number variation confounding persists at short physical distances.

Putting aside potential confounding due to physically proximal genetic perturbations, we observe other clear advantages to working

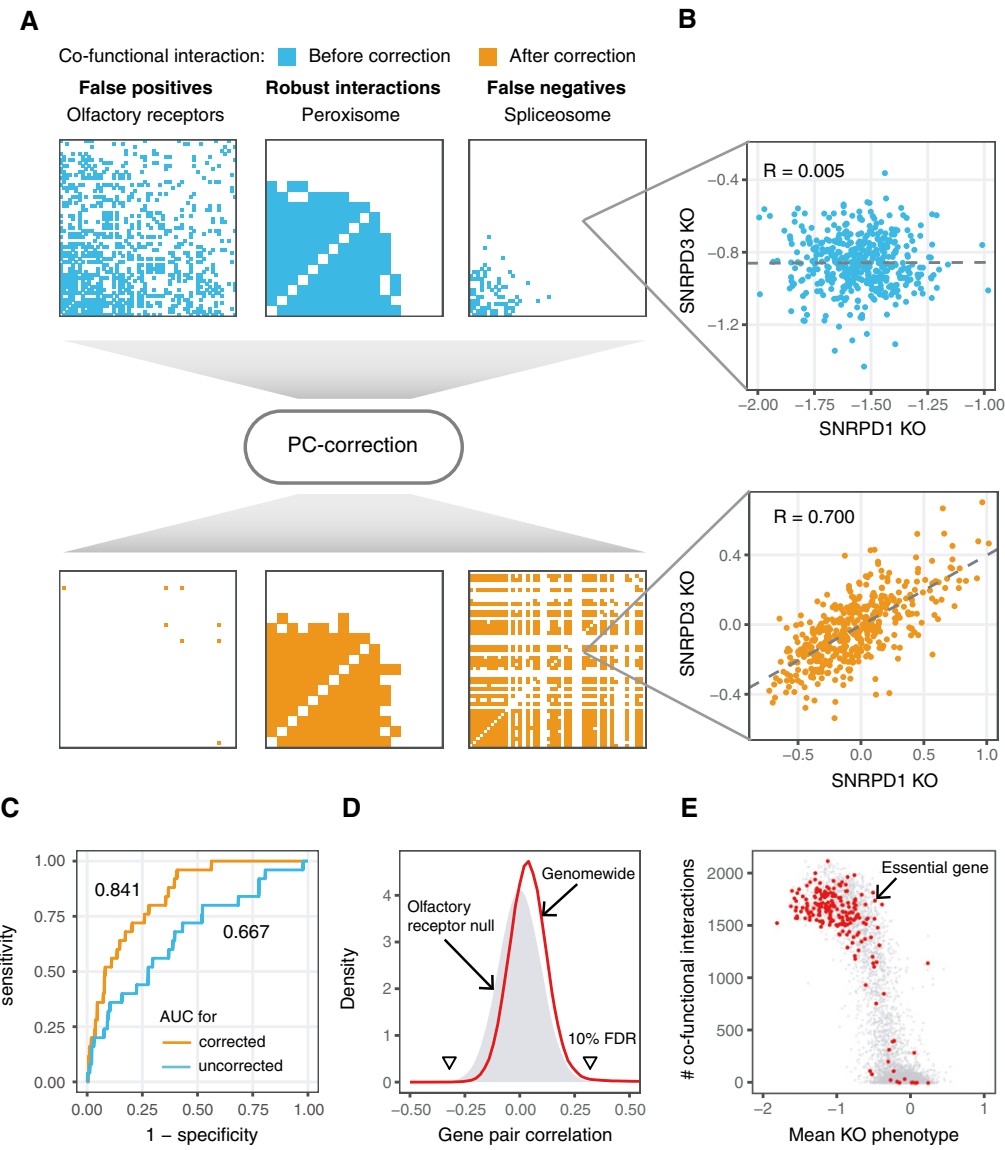

**Figure 2. Constructing a set of co-functional genes from all pairs of human genes.**

A   Demonstration of co-functional gene calls before and after correction for three example gene sets: olfactory receptors that exhibit nonspecific interactions in the raw data, peroxisome genes that show persistent co-functionality before and after correction, and spliceosome complex members that show correlation only after correction.

B   Correlation for two small nuclear ribonucleoproteins, *SNRPD1* and *SNRPD3*, that required for splicing, before and after correction of effect sizes from 436 cell lines. Fit lines are from linear regression.

C   Area under the receiver operating characteristic (ROC) curve for using cell line profiles to determine whether two cell lines share the same primary tissue of origin. Cell line profiles (the vector of effect sizes across all genes for each cell line) are more correlated among cancers of the same type following correction.

D   The distribution of pairwise correlations of corrected gene essentiality profiles genome-wide is greatly skewed to more positive values compared to pairs of olfactory receptors. Significance thresholds indicated with triangles.

E   Highly expressed essential genes regularly have on the order of one thousand co-functional interactions.

with corrected gene essentiality profiles. By creating cell line profiles from all gene knockout effects and comparing the correlations for every pair of cell lines before and after correction, we observed marked boosts in accuracy for predicting shared primary tissue (AUC increased from 0.667 to 0.841) and for predicting secondary tissue (AUC increased from 0.637 to 0.788; Figs 2C and EV1E). At the same time, the median cell line profile correlation dropped from ~0.85 to nearly 0 following correction (Fig EV1F), suggesting that

mean effect sizes per gene are lost in the correction process, although this does not affect calling of correlated gene pairs.

## Identification of over 1 million co-functional interactions from corrected gene essentiality profiles

To detect gene pairs with significantly correlated gene essentiality profiles, or "co-functional" genes, we derived *P*-values for the

observed correlations from an empirical null distribution. To build the distribution, we used the pairs of olfactory receptors described above as a background set, assuming that these receptors would not affect cancer growth. Correlations across all pairs of olfactory receptors were roughly normally distributed and centered at zero. Observed correlations across corrected gene essentiality profiles genome-wide for all other genes greatly exceeded expectation from the null model (Fig 2D). Thus, we assigned p-values for every observed correlation, whether positive or negative, using a normal distribution fit to the correlations of pairs of olfactory receptors. Across all 17,634 genes tested, 1,528,726 co-functional interactions, equal to 0.98% of all possible pairs of genes, met a false discovery rate of 10%. A similar procedure on raw gene essentiality profiles yielded only 30,761 interactions. Downsampling of the number of cell lines included in the analysis shows that more cell lines tighten the null distribution and increase power to discover co-functional interactions (Fig EV1G).

We immediately noted that highly expressed essential genes (Hart *et al*, 2014) generally possessed hundreds of co-functional interactions (Fig 2E). One might expect universally essential genes to lack identifiable co-functional gene partners, but this was rarely the case. Furthermore, we confirmed past reports (Barretina *et al*, 2012; Hart *et al*, 2015; Wang *et al*, 2015) that genes that are either highly or invariantly expressed across tissues on average possess greater knockout effects (Fig 3A). Remarkably, this holds true not only across cell types, but even in the context of expression data from a panel of lymphoblastoid cell lines (LCLs; Fig 3B; Pickrell *et al*, 2010), suggesting that broad mechanisms of gene regulation and not tissue specificity are responsible for the trend. As reported previously (Wang *et al*, 2015, 2017; Pan *et al*, 2018), differences in essentiality across genes thus appear quantitative and not strictly binary in each cell line. To facilitate exploration of our co-functional gene dataset, we have developed a shiny app that visualizes all genes co-functional to the query gene with added functionality for overlaying interactions from STRING, published CRISPR screens, or custom files containing scores per gene.

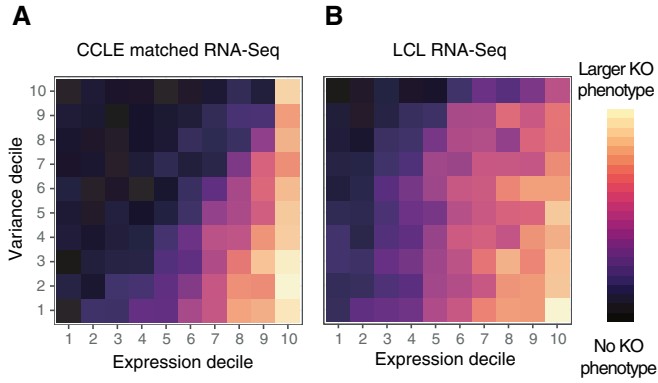

**Figure 3. Influence of RNA expression mean and variance on gene essentiality.**

A, B  Genes in high mean (*x*-axis) and low variance (*y*-axis) expression bins tend to have larger knockout effect sizes, whether expression is determined from (A) a matched RNA-Seq dataset or (B) a panel of LCL cell lines.

## Gene co-functionality captures variation in drug susceptibility across cancer cell lines

Past analyses (Hart *et al*, 2015; Wang *et al*, 2017) explored cell signaling in a restricted number of cancer cell lines (Fig 4A). With 436 samples, a much broader view of the diversity of signaling is possible, including differences across diverse cell types of origin. For MAPK and p53 pathways, the profiles of many genes hew closely to the *TP53* status of the cancer, producing two well-separated clusters with predictable effect sizes across cell lines (Fig 4B). Excluding a handful of outliers, *TP53* loss-of-function mutants (Barretina *et al*, 2012) do not respond to *TP53* or *MDM2* knockout or treatment with Nutlin-3, an anti-*MDM2* drug. *TP53* wild-type cells, in contrast, grow following *TP53* knockout, die following *MDM2* knockout, and exhibit slowed growth following treatment with Nutlin-3. These observations suggest that negative regulation of *TP53* is robust across cancer types, at least for its strongest co-functional genes. For example, two proposed drug targets, the deubiquitinase *USP7* (Gavory *et al*, 2018) and phosphatase *PPM1D* (Ogasawara *et al*, 2015), consistently mirror *TP53*-knockout phenotypes, proving themselves robust to cell lineage and mutations in other biological pathways.

The essentiality of BRAF and its co-functional gene partners paints a similar picture (Fig 4C), primarily with respect to melanoma cell lines. The knockout effects of genes co-functional to *BRAF* depend considerably on *BRAF* V600E status, with *BRAF* V600E lines especially sensitive to *BRAF*, *MAP2K1,* and *MAPK1* knockout, which another group independently reported (preprint: Kim *et al*, 2018). Thus, there is strong evidence that genes co-functional to central cancer growth genes mediate their essentiality through their involvement in those genes' pathways.

### Incorporating drug–gene associations into gene networks

By correlating the maximal activities of anticancer compounds in each cell line to the gene profiles derived from knockout effects, it is possible to test for drug–gene associations and then combine drugs and genes into a single network (Figs 4D and EV2). To explore examples of drug–gene interactions, we examine interactions with genes encoding the four ErbB proteins, well-characterized receptor tyrosine kinases (RTKs) that act upstream of PI3-K and MAPK pathways and complex with each other to mediate growth signaling (Medina & Goodin, 2008). ErbB family members are highly druggable and have been targeted by erlotinib, an anti-EGFR drug, and lapatinib, a dual anti-HER2, anti-EGFR drug. We observe that lapatinib-sensitive cell lines are sensitive to knockout ErbB family members (*EGFR*, *ERBB2*/HER2, and *ERBB3*/HER3), whereas erlotinib-sensitive lines are associated with sensitivity to *EGFR* and no other ErbB member.

For each drug compound, we tested the enrichment of KEGG terms among drug–gene associations. The most enriched term for each drug tended to confirm known biology (Fig 4E): Nutlin-3 upregulates p53 signaling, PLX4720 treats melanoma, and sorafenib modulates MAPK signaling. AZD6244 targets MAPK signaling pathways important for melanoma, explaining the extreme enrichment for "melanoma" genes. Enrichment for SNARE proteins for lapatinib associations suggests that RTK receptor

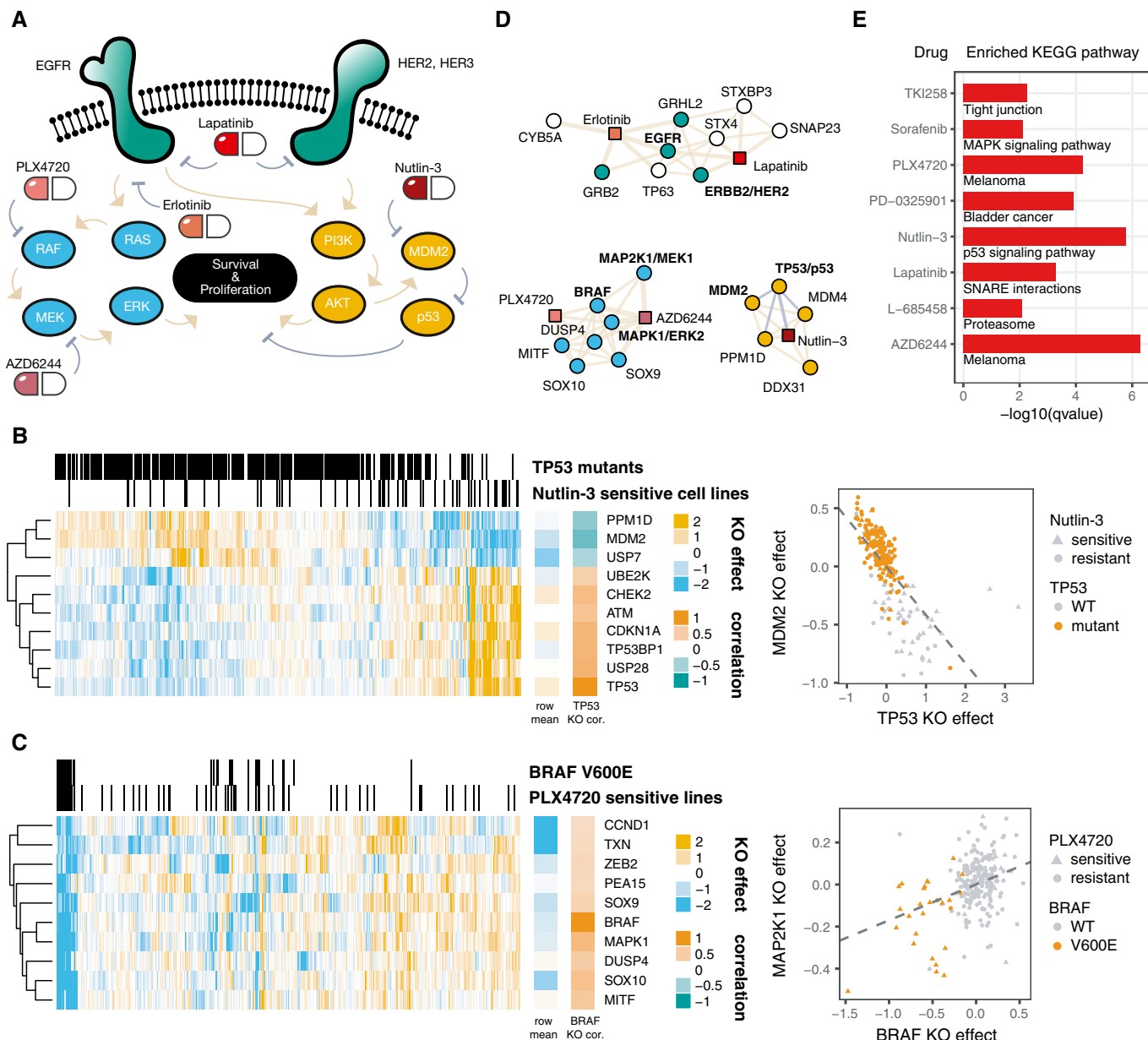

**Figure 4. Drug–gene associations in cancer cell networks.**

A   Schematized version of selected cancer growth signaling pathways: ErbB family members (green), MAP kinase (blue), and p53-Akt (gold).

B   Example of the most correlated genes co-functional to TP53. Each column represents one of 436 cell lines. Cell lines with TP53 loss-of-function (LOF) mutations or sensitivity to Nutlin-3 (binarized by *k*-means) are indicated in strips above. The mean effect for each gene is shown to the right of the heatmap, as is the corrected gene essentiality profile correlation with TP53. Effect sizes are variance-normalized by row. Scatterplot of the TP53 row vs. MDM2 row is shown on right.

C   Same as in (B) but for BRAF and its top co-functional genes.

D   Illustration of the top five drug–gene associations for each of the drugs in (A) plus their strongest co-functional genes. Beige edges represent positive correlations, gray negative.

E   Top KEGG pathway enriched for the top correlated genes of each drug shown.

trafficking is influencing the drug–gene network. Interestingly, L-685458, a gamma-secretase inhibitor, was most enriched for proteasome genes. There is debate in the literature whether gamma-secretase inhibitors, including L-685458, mediate their cancer-killing effects via gamma-secretase or by the proteasome (Clementz & Osipo, 2009). Cancer models often lack growth phenotypes when treated with gamma-secretase inhibitors, suggesting that these inhibitors hit multiple pathways such as the proteasome. In this panel, it is thus plausible that proteasomal inhibition is the most important factor.

## Enrichment of co-functional genes in curated gene set databases

To evaluate the extent to which co-functional genes reflect distinct kinds of functional relationships, we calculated the global enrichment of co-functional gene relationships across gene sets maintained by the Molecular Signatures Database (Subramanian *et al*, 2005). Curated gene sets consistently contained more co-functional pairs of genes than expected by chance: twofold enrichment for genes annotated with the same biological process or molecular function, more than threefold for cellular component, and fourfold for KEGG pathways (Fig 5A; Ashburner *et al*, 2000; The Gene Ontology Consortium, 2017). With respect to reconstituting curated gene sets, our co-functional gene dataset exhibits lower enrichment than protein mass spectrometry but contains over 100 times more potential interactions (Rolland *et al*, 2014). Approximately one-third of biological process and 60% of cellular component terms were

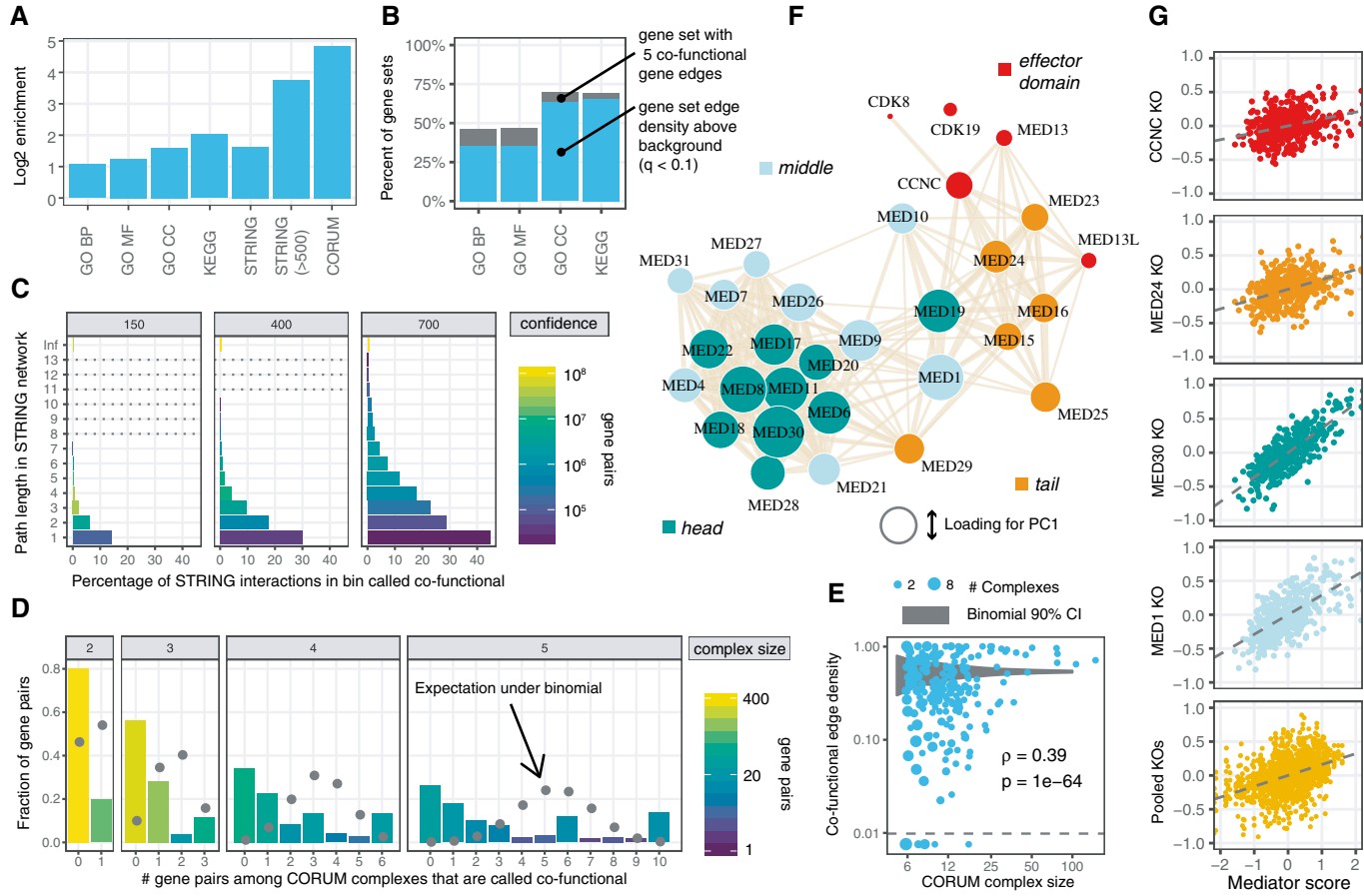

**Figure 5. Comparison of gene co-functionality to diverse gene annotation databases.**

A   Pairs of genes annotated with the same Gene Ontology term (BP = biological process, MF = molecular function, CC = cellular component) or Kyoto Encyclopedia of Genes and Genomes (KEGG) pathway are enriched for co-functional interactions. Pairs of genes with annotated interactions from STRING, especially high-confidence (> 500) interactions, are also enriched. Pairs of genes belonging to the same protein complex as curated in the Comprehensive Resource of Mammalian Protein Complex (CORUM) core complex database exhibit the greatest enrichment.

B   Percent of gene sets containing at least 5 pairs of co-functional genes and, among those, the fraction that are enriched for co-functional interactions above degree-matched random graphs.

C   Rates of co-functional gene calls for gene pairs binned by their path length in the STRING experimentally derived PPI network. "Inf" (infinite) refers to genes that lie in separate components.

D   Co-functional call rates among CORUM protein complex edges split by complex size. The binomial expectation for the number of edges called as co-functional is shown in gray. Co-functionality rates per complex are bimodal and greater for large protein complexes.

E   Extension of (D) to larger protein complexes, with the number of edges expressed as the fraction of all possible edges in the complex, equal to $\binom{n}{2}$. The area of each point reflects the number of multiple protein complexes with the same complex size and edge density. Complexes with fewer co-functional genes than expected according to the average rate are shown below the dashed line. The 90% binomial confidence interval for random co-functionality calls given the average rate is shown in gray. Large complexes again exhibit more co-functional interactions.

F   Reconstruction of the Mediator complex from gene co-functionality, shown in a force-directed graph. Subunits in the same domain (head, middle, tail, or CDK effector) of the complex are more likely to be co-functional than subunits in different domains. The area of each gene node corresponds to its loading on the first principal component of the corrected gene essentiality profile matrix.

G   Mediator subunit knockout effect sizes in each cell line can be summarized by a "Mediator score", the first principal component score for each cell line, plotted against the knockout phenotype of specific genes (first four facets) or a pool of all genes in the complex (bottom row).

enriched above levels seen for matched random networks (Fig 5B, see Materials and Methods). We also performed gene set enrichment on genes with no co-functional gene edges and found that underrepresented pathways were mostly limited to genes involved in lineage-specific (e.g., muscle) development and differentiation (Fig EV3A). Despite these blind spots, we conclude that many diverse biological pathways, not simply pathways typically associated with essentiality, contribute to gene co-functionality.

### Co-functional gene edges overrepresented among protein–protein interactions

Relative to curated gene sets, reported protein–protein interactions exhibited greater enrichment of co-functional genes. This was true for large-scale repositories of experimental data, as curated by the STRING consortium (Szklarczyk *et al*, 2015) (14-fold among interactions above 500 confidence), and for well-studied protein complexes, as curated in the CORUM core complex resource (Ruepp *et al*, 2010) (almost 30-fold for proteins in the same complex). In fact, we validated the majority (53.9%) of gene pairs in CORUM core complexes ($N = 48,408$) as co-functional genes (binomial test $P < 1e\text{-}300$).

We next explored how STRING interactions related to our set of co-functional genes in more detail. 13% of co-functional interactions were present in the STRING database of protein–protein interactions, compared to 3% of all pairs of genes. The probability of a STRING interaction being called varied depending on the type of annotation and in accordance with its confidence (Fig EV3B). 11% of low-confidence (score 1–150) experimentally derived STRING interactions were among our co-functional interactions, compared to 88% of high-confidence (score 900-1000) STRING interactions. Likewise, only 4% of low-confidence co-expression transferred STRING interactions were among our co-functional interactions, compared to 72% of high-confidence STRING interactions. Nonetheless, all categories were enriched for co-functional genes above the global average of 0.98%.

Interestingly, we find only a small fraction (~2%) of co-functional genes to be negatively correlated. Among pairs of genes in STRING with detailed functional annotations, 32% of the highest confidence (> 900) activating interactions were called, in contrast to 7.8% of highest confidence inhibiting interactions (Fig EV3C). High-confidence (score > 700) activation, catalysis, reaction, and binding interactions were overwhelmingly positively correlated (99.4%, 99.3%, 99.4%, and 99.5% of correlations above zero, respectively, Fig EV3D). While the statistically significant correlations among inhibiting interactions might be expected to be broadly negative, this was not the case: Only 1.8% of all correlations were negative. We found that 61% of STRING interactions annotated as inhibiting were also annotated as activating or catalyzing and could predominantly operate in a cooperative rather than inhibitory manner, but even among pairs of genes exclusively annotated as sharing an inhibitory relationship, only 36% were negatively correlated. This dearth of anti-correlated co-functional genes is consistent with the observation by Wang *et al* (2017) that anti-correlated co-essential genes are less often detected. Thus, pooled CRISPR-knockout screens, even with diverse panels of cell lines, may have low sensitivity for detecting inhibitory relationships across pairs of genes, or negative regulation more broadly.

Because our reported number of co-functional interactions exceed the estimated number of human gene pairs that physically interact (Stumpf *et al*, 2008; Venkatesan *et al*, 2009) by as much as 10-fold, we explored the potential underpinnings of co-functionality signal in greater detail. In total, only 12% of co-functional genes are annotated by any co-expression/experimental study, suggesting that physical interactions among proteins and gene co-regulation explain a minority of co-functional interactions. To evaluate the extent to which co-functional genes could be linked indirectly by cascades through the human interactome, we calculated the rate of gene co-functionality as a function of the path length dictated by the STRING database. Pairs of genes separated by 2–4 experimentally derived STRING edges had consistently higher rates of co-functionality compared to pairs of genes in separate components of the network. Among STRING interactions of confidence > 700, genes with one intermediate gene, or path length 2, were enriched 30-fold (Fig 5C). The functional pathways we identify are likely mediated in part by intervening protein interactors, in additional to metabolic pathways and direct physical interactions.

### Characterizing co-functionality among members of protein complexes

Across CORUM core complexes, we also observe that the number of co-functional interactions per complex deviates far from expected assuming independent draws from a binomial distribution. In fact, oftentimes all or none of a protein complex's members were labeled co-functional (Fig 5D and E). Complex size also played a role, as 60% of core complexes with 5 or fewer members contained no co-functional genes, whereas 95% of complexes with more than five members exceeded the average rate of co-functional interaction calls. Protein complex essentiality appears to explain some of this bimodality: Co-functionality is rarely detected among CORUM core complex members that only weakly affect growth, whereas CORUM core complexes that strongly affect growth form nearly complete graphs of co-functionality (Fig EV3E), as seen for mitochondrial and cytosolic ribosomes, the proteasome, U2 snRNP, and RNA polymerase II complexes.

Although the rate at which we identified the members of a protein complex co-functional varied according to the knockout growth phenotype, we identified other factors that guided which of a complex's members were co-essential and which were not. The Mediator complex, a transcriptional coactivator that has been previously examined using a similar parallel screening analytical approach (Pan *et al*, 2018), serves as an example. Although knockout of every Mediator complex member is associated with impaired growth, we called fewer than half of the pairs of genes comprising the Mediator complex co-functional (Fig 5F). The network of co-functional gene edges describing the members of the Mediator complex in fact mirrors the three-dimensional structure and function of the complex; gene knockouts cluster by their topology, divided into head, middle, tail, and effector module (Yin & Wang, 2014). We also summarized cell lines by running PCA on the matrix of cell lines by Mediator complex members and assigning the first PC score of each cell line as its "Mediator score". This score integrates the 30 Mediator member knockout effect sizes into a single number reflecting the growth defect associated with Mediator loss of function. The Mediator score can

predict the effect of knocking out arbitrary Mediator complex members (Fig 5G) but does so independently of protein domain-specific effects.

## Gene communities in the cancer cell network offer insight into cancer proliferative processes

To examine the distribution of co-functionality genome-wide and nominate candidate core genes for cancer growth, we performed community detection to partition all genes into separate communities (Fortunato, 2010). Every gene was assigned to one of 2,857 separate communities, including 2,978 singleton genes with no co-functional genes and 562 small communities with fewer than 8 genes. Because singleton and very small gene communities harbor few edges and are better suited to gene- and not community-centric approaches, we focused on the 326 communities with at least 8 gene members (Fig 6A).

We first enumerated the 719 COSMIC census genes in each community (Forbes *et al*, 2017) and found that over one-third of gene communities (126/326) contained at least one census gene, illustrating the extreme diversity of pathways that cancers can manipulate to maximize cellular proliferation. However, certain gene communities possessed many more census genes than expected, including communities containing *TP53*, *EGFR*, *KRAS*, and *BRAF*. Finally, over half of the clusters (206/326) were significantly enriched for at least one Biological Process term or KEGG pathway at a false discovery rate of 10%.

The largest community we identified was densely connected (12% of all pairs connected) and encompassed numerous core essential processes identified by KEGG, including the spliceosome, the ribosome, the proteasome, cell cycle, RNA polymerase, and mismatch repair (Fig 6B). A smaller, slightly denser cluster (13% of all pairs connected) contained genes required for mitochondrial function and aerobic respiration. Genes lacking any pathway annotations for the most part were not enriched in any gene communities but were significantly dis-enriched in these core growth communities (Fig EV4).

Gene communities varied substantially in the breadth of their encapsulated functions. Some communities derived from specific cancer pathways, such as the community containing *EGFR*, which simultaneously captured signaling from related ErbB proteins and the role of cell adhesion and tissue migration in *EGFR*-dependent transformation (Fig 6C) (Lindsey & Langhans, 2015). Other clusters reflected very specific cell functions or compartments, as in the case of the peroxisome.

In the peroxisome community, 34 Gene Ontology terms were statistically enriched, ranging in size from 12 to 415 genes. Enriched gene sets primarily derived from peroxisomal transport and fatty acid metabolism. Although these are not conceptually similar pathways, they both relate to the core function of peroxisomes, which are variably essential across the cancer cell lines. In the context of cancer proliferation, abstracting fatty acid metabolism away from peroxisomal transport genes, as done in the Gene Ontology database, may ignore the reality that these processes are functionally inseparable.

Another community (Fig 6D) contains the genes underlying hereditary multiple osteochondromas, a rare Mendelian disorder. These genes, *EXT1* and *EXT2*, are known to act in the

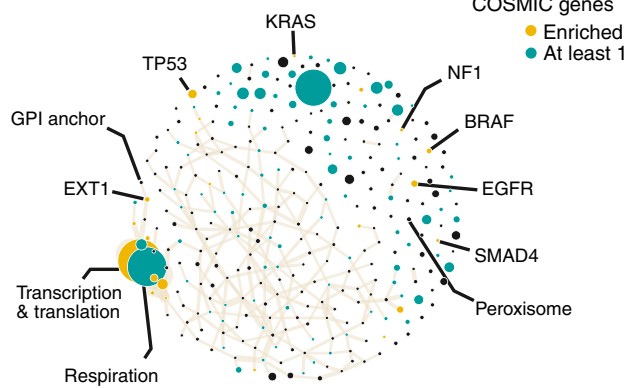

**A**  Global network of gene communities

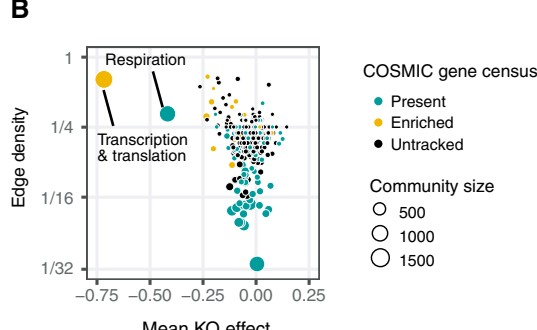

**B**

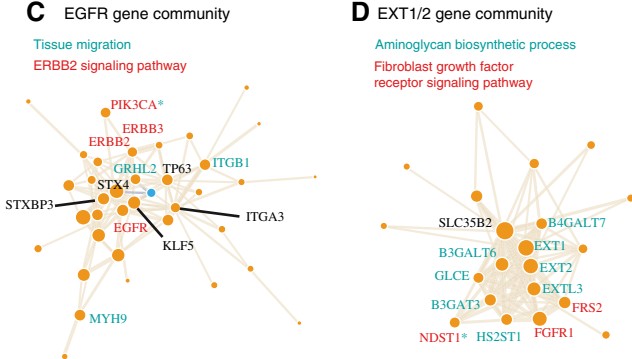

**C**  EGFR gene community

**D**  EXT1/2 gene community

---

**Figure 6. Gene communities and network topology in cancer cell networks.**

A    Consolidation of genes into > 200 communities using de novo community detection. COSMIC census genes frequently cluster in the same gene communities beyond expected by chance (in gold) but are otherwise widely dispersed throughout the network (in green). Nodes representing communities are sized by number of constituent genes. The largest community contains over 1,000 core essential genes (labeled "Transcription & translation"). Select gene communities are labeled by enriched annotations and/or prominent cancer-associated genes.

B    Distribution of gene communities by mean knockout effect size of community members and co-functional edge density. "Respiration" and "Transcription & translation" from (A) are both broadly essential and densely connected.

C, D  Depictions of the *EGFR* and *EXT1* gene communities. Size of the gene nodes reflects loading on the first principal component among community members. Gene symbols that are members of selected enriched gene ontology terms are labeled.

polymerization of the glycosaminoglycan heparan sulfate, a known cofactor for FGF signaling (Ornitz & Itoh, 2015). How *EXT1* loss leads to cancer is poorly understood (Bovée, 2008); nonetheless, eight other known aminoglycan synthesis genes participate in the same gene community. Also present are fibroblast growth signaling genes *FGFR1*, *FRS2*, and *NDST1*. *GRB2* and *PTPN11* are not in the same gene community but are linked to gene community members by additional co-functional interactions. One possibility is that every member of the aminoglycan synthesis pathway influences cancer proliferation by ultimately making heparan sulfate available to upregulate FGF pathways. If true, the number of genes that could potentially modify cancer proliferation via FGF signaling is much larger than currently appreciated, expanding from *EXT1* and *EXT2* to all aminoglycan synthesis genes.

### Gene network centrality adds another layer to gene function

We next examined the network topology within and across gene communities. Because core growth genes would dominate strength or degree calculations for most genes, we quantified the centrality of every gene using the closeness of each gene within its prescribed community. The closeness of a gene is the reciprocal of the sum of the shortest distance via co-functional gene edges from that gene to every other gene in the network. By calculating this metric within communities, we obtain a local centrality measure that gives insight into a diverse range of gene functions.

As a measure of centrality in the cell network, closeness added information not captured by other gene properties such as essentiality or expression. Overall, COSMIC census genes did not differ in closeness compared to other genes (Wilcoxon rank-sum $P > 0.05$), nor did they differ substantially in their average knockout growth phenotype. However, genes annotated as germline cancer genes scored much higher in closeness than somatic cancer genes (Fig 7A, Wilcoxon rank-sum $P = 7e-9$). In fact, gene closeness surpassed gene expression level or knockout growth phenotype in accuracy for ascertaining somatic from germline cancer genes (Fig 7B). Closeness was slightly greater for genes suspected to be subject to strong purifying selection via population genetic data (high pLI genes; Lek *et al*, 2016), but high RNA expression level was more predictive of high pLI status than closeness (Fig EV5A). We also observed that genes linked to unfavorable prognoses in cancer patients exhibited greater closeness in gene communities, but that genes linked to favorable prognoses did not (Fig 7C; Uhlen *et al*, 2017). In all, we linked high closeness genes to oncogenes and germline cancer processes, complementing broad patterns of high expression and essentiality for cancer drivers.

To summarize cancers by gene community activity, we again performed principal components analysis—this time separately for each gene community (see Materials and Methods)—and recorded the first PC score for each cell line. This score reduces the dimensionality of the gene profiles from the size of the gene community down to a single number, which we term the community score. One benefit of scoring communities as opposed to individual genes is that the shared function of a gene community can be ascertained while avoiding the measurement noise or alternate signals associated with individual gene knockouts.

We first explored how community score varied by cell lineage and confirmed that cancer cell lineages dependent on particular

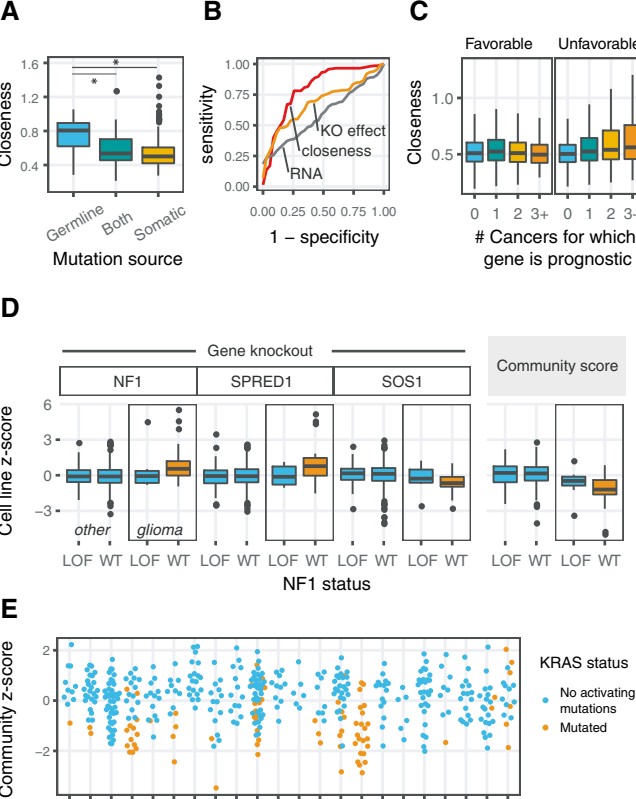

**Figure 7.  Topology and cell type specificity of cancer networks.**

A   Boxplot of closeness for genes annotated as germline cancer genes, somatic cancer genes, or both. Significant by Wilcoxon rank-sum test. Upper comparison *$P = 7e-9$; lower comparison *$P = 7e-4$.

B   ROC curve showing how centrality (measured by closeness in the gene community) separates germline from somatic cancer genes in the COSMIC gene census more accurately than either growth knockout phenotype or expression.

C   Genes associated with unfavorable prognoses as found in the Human Protein Atlas Pathology database have greater closeness within gene communities, left. The same is not true for genes associated with favorable prognoses, right.

D   NF1 gene community knockout phenotypes and the NF1 aggregate community score calculated from the first principal component of member genes shows that perturbation of the community by CRISPR knockout almost always occurs in NF1-WT glioblastoma cell lines but only rarely in other cancers. $P < 0.05$ for all within-glioma comparisons.

E   KRAS community scores demonstrate that dependence on constitutively active (mutant) KRAS drives community organization, independent of tissue, but is uniformly present across pancreatic cancers.

Data information: (A, C, D) Boxes reflect IQR, and lines reflect 1.5 times the IQR. There are no replicates. Data annotations of cell type are as described in the Meyers *et al* data depository (Meyers *et al* 2017; Data ref: Meyers *et al* 2017).

growth pathways were associated with extreme scores for the corresponding gene communities. For example, it was often possible to classify the lineage of glioblastoma, neuroblastoma, small-cell lung cancer, melanoma, and pancreatic- and kidney-derived cancers by

virtue of their especially extreme community scores (Fig EV5B), analogous to others' findings (preprint: Kim *et al*, 2018). Interestingly, we found that communities associated with cell lineage identity (see Materials and Methods) were three times more likely to be enriched for COSMIC census genes than gene communities with no associations (Fisher's exact test $P$ = 2e-8), consistent with the interpretation that cancers from different cell lineages target different growth pathways to maximize growth.

### Diversity of cell line panel expands the scope of detectable cell signaling paradigms

In some cases, cell lines of a specific lineage and mutational background were required to uncover cancer-relevant growth pathways. The gene community containing the tumor suppressors *NF1* and *SPRED1* illustrates this phenomenon (Fig EV5C). In the germline, one *NF1* loss-of-function allele causes the Mendelian disorder neurofibromatosis type 1 (Gutmann *et al*, 2017). Similarly, loss of function of *SPRED1* causes Legius syndrome, which can be confused clinically with neurofibromatosis type 1 (both genes act upstream of RAS signaling). In the panel of cell lines from Project Achilles, glioma cell lines consistently exhibit the largest effect sizes for community members *NF1*, *SPRED1,* and *SOS1*. Yet, even among glioma cell lines, only those that lack a preexisting *NF1* loss-of-function event exhibit large differences in sensitivity to gene knockout of community members (Fig 7D). The differences across these genes are well summarized by *NF1* community scores, where glioma cell lines with intact *NF1* consistently score above glioma cell lines with *NF1* loss of function (Wilcoxon rank-sum $P$ = 0.005). In the case of non-glioma cell lines, NF1 community scores rarely deviate from the mean and do not vary by *NF1* mutation status (Wilcoxon rank-sum $P$ > 0.05).

While extreme community scores often occur in distinct cell lineages, sensitivity to perturbation of gene communities can often be accessed across multiple lineages. For example, perturbations of the *KRAS* gene community are strongly associated with strong fitness effects in pancreatic and colorectal cancers, but also exhibit apparent effects in cell lines derived from other lineages. In general, extreme *KRAS* community scores indicate the presence of mutant, constitutively active *KRAS* (Fig 7E), with pancreatic and colorectal cancers very likely to acquire such mutations due to inherent tissue-specific cancer biology. In the Project Achilles panel, all pancreatic cancers harbor *KRAS*-activating mutations, while ovarian cancers infrequently do. This difference in mutational status explains the difference in *KRAS* community scores between pancreatic and ovarian cancers more accurately than cell lineage and stands in contrast to the *NF1* gene community example where both a specific lineage and mutational status were important characteristics.

## Discussion

This work demonstrates the power of unsupervised statistical techniques for correcting gene profiles constructed from parallel screening datasets. We show that technical confounding is pervasive across CRISPR-knockout screens, but that highly active sgRNA libraries and extensive data preprocessing steps can expand the

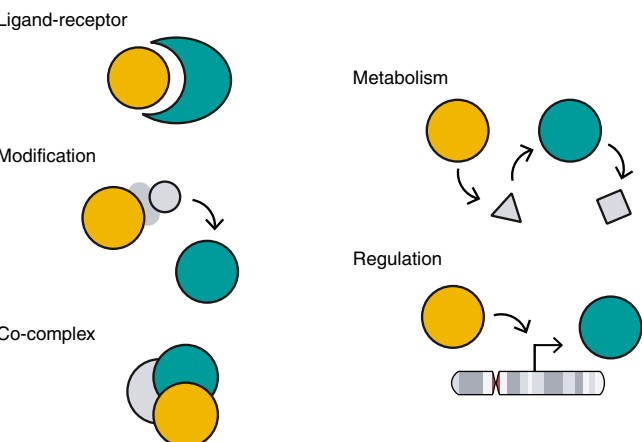

**Figure 8. Co-functional gene relationships.**
Co-functionality learned from correlations among gene profiles broadly extends to gene–gene relationships from physical interactions like ligand–receptor interactions, co-complex formation, or post-translational modification to conceptual relationships like gene regulation and shared metabolic pathways.

quantity and quality of interactions called from correlated gene profiles, whether or not interactions are mediated directly by protein–protein interactions (Fig 8). When screening for hit genes that modify a phenotype of interest, it is already considered best practice for sgRNA libraries to contain "safe-targeting" sgRNAs that target non-genic regions to correct for the toxic effects of DNA cleavage (Morgens *et al*, 2017). We provide evidence that sgRNA off-target effects can cause both false-positive and false-negative interactions, and, by exploiting a set of control genes with a very low prior of having an effect on the screen phenotype, we can correct these confounding signatures. In the future, more comprehensive sets of control sgRNAs may further improve modeling of confounding across genomic regions of variable copy number, sequence content, and chromatin accessibility.

Although some biological pathways can be readily detected using a relatively small number of genetic screens for cancer growth, a comprehensive mapping of cell networks will require much more diverse panels of screens. In multiple cases, we detected certain pathways that were active only in specific cell lineages or mutational backgrounds. Among screens currently available, inhibitory relationships among genes are only rarely detected. Finally, gene knockouts that lack a growth phenotype in the available cell lines cannot be incorporated into cell networks at all. These findings argue for collecting screen data (a) for samples of cells with diverse mutational and cell lineage backgrounds and (b) across diverse screening conditions. The use of activating (CRISPRa) screens might also improve detection of antagonistic gene pairs. Parallel screens in the same cell lines for cancer phenotypes other than growth, such as invasiveness or cell size, would complement existing data on proliferation, but screening other phenotypes, such as phagocytosis or sensitivity to oxidative stress, offers a way to improve the richness of sparse gene profiles. As more screening data become available that differ by screening phenotype, laboratory, manner of perturbation (CRISPRi/a, base editing, inducible systems), and sgRNA library, quality control for parallel screening techniques will become increasingly critical.

Maps of cancer cell networks drawn using co-functional interactions are assuredly detailed, but parallel screening approaches come with distinct limitations. First, the co-functional interactions we report obscure the distinction between direct and indirect interactions as well as the directionality of information flow in signaling processes. De novo methods to call gene communities might cluster genes into a smaller set of modules, but the manner by which genes in the same module work together remains difficult to infer. One open question specific to the field of parallel screening is whether interactions identified by double-knockout screens, particularly synthetic interactions, can be predicted from co-functional interactions identified in data collected from other cell lines. Also unknown is the extent to which gene communities learned from one phenotype are the same communities underlying other phenotypes. Yet, information theoretic models and orthogonal genome-wide profiling data (e.g., RNA-Seq, ATAC-seq) promise to expand what can be learned from genome-wide perturbation data. Regardless, engineering additional pre- and post-processing computational methods for genome-wide perturbation data are likely to advance progress toward a comprehensive understanding of cell network logic.

# Materials and Methods

### Calling co-functional interactions from CRISPR screening data

Project Achilles gene-level effect sizes (version 18Q2), the lists of 217 highly expressed genes and 927 nonessential genes from Hart *et al*, and sgRNA sequences were downloaded from Meyers *et al*'s data depository (Meyers *et al*, 2017a; Data ref: Meyers *et al*, 2017b) for further processing.

To correct cell line-specific Cas9 toxicity, nonspecific cell line signatures were generated from olfactory receptor gene essentiality profiles. The HGNC olfactory receptor gene list was downloaded from the HGNC website (https://www.genenames.org/cgi-bin/genefamilies/set/141). The full list of olfactory receptor gene symbols was intersected with gene symbols present in the gene-level statistics. Some sgRNA sequences were not unique among olfactory receptors; in these cases, one olfactory receptor was selected at random and any olfactory receptors with duplicate sgRNA were discarded. The resulting matrix of 250 olfactory receptors by 436 cell lines was transposed and subjected to PCA using the *prcomp* function in R. The same procedure was applied to 100 permutations of the same matrix by shuffling effect sizes within columns (cell lines). For the first five PCs, the proportion of variance explained per PC for the true matrix exceeded that of all permutations and were deemed signatures of nonspecific toxicity.

Gene essentiality profiles were projected onto the principal components identified as nonspecific and converted back to the original dimension via matrix multiplication, and the difference of matrices was taken as a set of corrected gene essentiality profiles.

In contrast to the original, frequently positively correlated gene essentiality profiles, correlations among corrected gene essentiality profiles for olfactory receptors resembled a normal distribution centered at zero. To establish a null distribution, the above steps were repeated with fivefold cross-validation and the mean squared correlation of olfactory receptor gene pairs across all folds was calculated as the standard deviation, with mean 0 (Fig 2D). This

process was repeated with curated nonessential genes and yielded a 22% narrower standard deviation. *P*-values were assigned to all human gene pairs using this normal distribution via the *pnorm* function in R, and hits at a 10% FDR were identified as co-functional interactions.

Using the same correlation cutoff for gene essentiality profiles, co-essential gene hits were compared to co-functional genes within three sets of genes: Gene Ontology "spliceosomal complex assembly" genes (a category with surprisingly few correlated gene essentiality profiles), Gene Ontology "peroxisomal transport" genes (a well-circumscribed set of co-essential genes), and olfactory receptor genes with co-functional degree of 12 or greater. In general, the number of co-functional interactions, or degree, varied considerably across all genes. Highly expressed essential genes were confirmed as being greatly enriched among genes with very high degree (Fig 2E).

### Predicting shared lineage and primary disease from cell line signatures

Cell line profiles were taken from the transpose of the corrected gene essentiality profile matrix, and pairs of cell lines were labeled as deriving or not deriving from the same cell lineage. This created 436 choose 2 observations of 0 (different primary tissue) and 1 (same primary tissue). The same was done for secondary tissue. ROC plots and AUC values on using correlation between cell line profiles to detect lineage or primary disease identity were calculated using the ROCR R package. The higher the accuracy, the better gene profiles are able to expose differences across cell types in biological pathway dependence.

### TP53 and BRAF co-functionality heatmaps

Germline-filtered mutation data, such as for TP53 and BRAF, and drug sensitivity data were downloaded from the CCLE portal (https://portals.broadinstitute.org/ccle/data). For TP53, all cell lines with a protein-coding or splice mutation were labeled loss-of-function mutants. For BRAF, cell lines with a V600E protein change were labeled V600E lines. Binarized drug sensitivity to either Nutlin-3 or PLX4720 followed from *k*-means clustering on the Amax values with *k* set to 2 to separate two groups of resistant and sensitive.

The top 9 genes co-functional to either TP53 or BRAF were visualized in the heatmap using the R package ComplexHeatmap. Corrected gene essentiality profiles (rows) were divided by their corresponding standard deviation to normalize the heat scale. The distance metric for genes (rows) was magnitude of Pearson's correlation. For cell lines (columns), signed Pearson's correlation was used. Accompanying scatterplots show the unnormalized gene knockout effect sizes with a linear fit trend line.

### Drug–gene network and associations

The reported Amax values were used as measures of drug efficacy for all cell lines. Drug phenotypes were then correlated with gene knockout effect sizes for all drugs across all genes. For network visualization (Fig 4D), the four genes with the largest magnitude correlation in either direction were selected. For drug–gene

association analysis (Fig 4E), the 30 most correlated gene profiles were taken for KEGG pathway enrichment for each compound. Only compounds with at least one enriched pathway were visualized.

## Gene set enrichment comparisons

Curated gene sets derived from Gene Ontology and the Kyoto Encyclopedia of Genes and Genomes were downloaded from the MSigDB website, version 6.1 (https://software.broadinstitute.org/gsea/msigdb/). All pairs of human genes were assessed for co-annotation by any ontology term. Aggregate enrichment of functional genes for these co-annotated gene pairs (Fig 5A) was calculated as

$$\log_2((e_{fc}/e_c)/(e_f/e))$$

where $e_{fc}$ is the number of gene pairs that are both co-functional and co-annotated, $e_c$ is the number of gene pairs that are co-annotated, $e_f$ is the number of gene pairs that are co-functional, and $e$ is the number of total gene pairs.

Several gene sets consist primarily of essential genes with large numbers of co-functional gene edges. Under these conditions, it is possible for a small number of genes or pathways with a large enrichment of co-functional gene edges to underlie a genome-wide enrichment, even if most pathways contain few or no co-functional genes. To estimate the number of gene sets contributing to the aggregate enrichment, we first filtered out gene sets with under 5 co-functional gene edges. These low-signal, sparsely connected gene sets are in some cases statistically enriched but would have limited use in interpreting or modeling of genome-wide networks. The remaining gene sets spanned slightly under half of GO biological process and molecular function gene sets and approximately 70% of GO cellular component and KEGG pathway sets.

Genes in the dataset were then placed into one of 100 bins based on their degree in the co-functional gene network. For each gene set, new, random gene sets were constructed such that each gene was replaced by a random gene from the same degree bin. The number of edges in the corresponding random subgraphs of the genome-wide network was calculated to serve as an empirical null distribution. If the random gene sets never reached the number of edges seen in a true gene set, a *P*-value of 0.5/(# permutations) was assigned to the gene set. Significantly enriched gene sets were called at a false discovery rate of 5%. We found that over 75% of the remaining GO biological process and molecular function terms exhibited more edges than expected, and more than 90% of the remaining GO cellular component and KEGG pathways.

## Genetic interaction and protein complex comparisons

Protein–protein and genetic interaction data were downloaded from the STRING v10.5 website (detailed and action files, https://string-db.org/cgi/download.pl). Distances between gene pairs were calculated using the *distances* function from the igraph R package. The rate at which co-functional interactions were called per path length was calculated as (# co-functional interactions) divided by (# total gene pairs).

CORUM core complexes were downloaded from the CORUM website (http://mips.helmholtz-muenchen.de/corum/#download). A dataset of protein complex edges was created by enumerating all pairs of genes that were members of the same human protein complex. Among all such edges, 53.9% were called as co-functional. The expected binomial distribution per complex (Fig 5D and E) was calculated using the *dbinom* function in R, and 90% confidence intervals were calculated using *qbinom*.

To calculate Mediator gene loadings and cell line PC scores, PCA was performed on the corrected gene essentiality profiles of Mediator complex members across 436 cell lines with the *prcomp* function in R. With genes as features, the mean effect size of each gene was subtracted to estimate the covariance across genes. All loadings for the first principal component were positive, meaning all Mediator complex members covaried in the same direction, and the magnitude of the loading was taken as the relative weight for estimating the function of the entire complex.

## Community-centric analysis of the cancer cell network

Co-functional gene edges were analyzed using the igraph package in R. Distances between two genes a and b were weighted by 1 − abs(cor(gene a, gene b)), and the edge width was scaled to abs(cor(gene a, gene b)). Communities were called using the *cluster_infomap* function. The edge density of each community was calculated by constructing a subgraph from the community members and calling the *edge_density* function. In order to prevent core essential genes from influencing measures of centrality, closeness was calculated for each gene locally by calling the *closeness* function on the community subgraphs. The gene community network plot (Fig 6A) was visualized by creating a new graph of communities as nodes. Node area was made proportional to the number of genes in the community by scaling the size parameter to the square root of the number of genes. For gene communities with greater than 100 genes, the community was discarded if more than 50% of the community's genes laid on the same chromosome. Edges were drawn between communities if the edge frequency between them surpassed the genome-wide average of 0.98% with width scaled to edge frequency.

Communities were annotated both by the number of members that were COSMIC cancer census genes (https://cancer.sanger.ac.uk/census) and by the enrichment in KEGG and Gene Ontology gene sets as determined by the ClusterProfiler R package. Enrichment of COSMIC census genes was determined by binomial test with probability of success equal to the total number of COSMIC census genes divided by the number of genes in the network and a false discovery rate cutoff of 10%. Enrichment of uncharacterized genes in gene clusters was determined similarly (Fig EV4), where an uncharacterized gene was defined as any gene lacking a Gene Ontology biological process annotation. To understand which biological processes were sparsely connected and poorly represented (Fig EV3A), communities with fewer than 8 members were analyzed using gene set enrichment.

The importance of the local closeness measure was first evaluated using the Human Protein Atlas pathology data (https://www.proteinatlas.org/about/download). For both favorable and unfavorable prognoses, genes were divided into four tiers: prognostic for 0, 1, 2, or 3+ cancer types. The Wilcoxon rank-sum test was used to compare local closeness across tiers (Fig 7C). How probability of loss-of-function intolerance (pLI) scores varied by closeness was also tested (http://exac.broadinstitute.org/downloads, file "fordist_cleaned_exac_r03_

march16_z_pli_rec_null_data.txt"), but was weakly predictive compared to gene expression (Fig EV4A).

### Gene community scores

Gene communities were processed by performing PCA on the transpose of the corrected gene essentiality profile matrix and recording the first principal component, as performed on the Mediator complex. Cell lines were scored by their first principal component score. Cell lines with large scores are interpreted as the drivers of the gene community. To examine cancer lineage and mutation status contributions to the *NF1* gene community (Fig 7D), community scores were aggregated by whether the cell lineage was glioma and whether the cell line possessed a non-silent mutation in *NF1*. To examine cancer lineage and mutation status contribution to the *KRAS* gene community (Fig 7E), community scores were aggregated by every lineage with more than 5 samples and whether the cell line possessed a missense mutation in *KRAS*. Significance of cell lineage contributions across all communities (Fig EV5B) was determined by permuting cell lineage labels and calculating mean community scores per cell lineage to generate a null distribution. Cell lineages that had more extreme scores than expected were called at a 5% FDR cutoff. The gene communities with the thirty most significant associations were visualized.

### Co-functional interaction examples

The chromosome idiogram in the gene regulation example was downloaded from the Human Genome Idiogram Vector Art library (https://github.com/RCollins13/HumanIdiogramLibrary). The chromosome 17 idiogram ai file was chosen for illustration.

## Data availability

Co-functional interactions are available at: https://greenleaf.shinyapps.io/cofunctional_app_18Q4. The Shiny app code and full co-functional interaction dataset with gene community information can be downloaded from FigShare: https://figshare.com/s/35a82ed1e48d0ec4e9e4.

**Expanded View** for this article is available online.

## Acknowledgements
We thank Michael Haney and Michael Bassik for advice on CRISPR screen data analysis. We also thank Eilon Sharon, Emily Glassberg, Nasa Sinnott-Armstrong, and other Pritchard and Greenleaf lab members for helpful discussions. This work was supported by the NIH (P50HG007735, RO1 HG008140, 1UM1HG009436), the Rita Allen Foundation, and the Human Frontiers Science Program grant (RGY006S). Evan Boyle is supported by the National Science Foundation Graduate Research Fellowship. William Greenleaf is an investigator of the Chan Zuckerberg Biohub. Jonathan Pritchard is an investigator of the Howard Hughes Medical Institute.

## Author contributions
EAB conceived the project, performed analyses, and drafted the manuscript. JKP and WJG supervised analyses. All authors edited and revised the text.

## Conflict of interest
The authors declare that they have no conflict of interest.

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
