## [Review Process File · Molecular Systems Biology]

High-resolution mapping of cancer cell networks using co-functional interactions

Evan A. Boyle, Jonathan K. Pritchard, William J. Greenleaf

Review timeline:

Submission date:	6 August 2018
Editorial Decision:	25 September 2018
Revision received:	26 October 2018
Editorial Decision:	22 November 2018
Revision received:	26 November 2018
Accepted:	30 November 2018

Editor: Maria Polychronidou

Transaction Report:

1st Editorial Decision

25 September 2018

Thank you again for submitting your work to Molecular Systems Biology. We have now heard back from the three referees who agreed to evaluate your study. As you will see below, the reviewers think that the presented findings seem relevant for the field. They raise however a series of concerns, which we would ask you to address in a major revision.

I think that the recommendations of the reviewers are quite clear, so there is no need to repeat the points listed below. Please do not hesitate to contact me in case you would like to further discuss any of the issues raised.

If you feel you can satisfactorily deal with these points and those listed by the referees, you may wish to submit a revised version of your manuscript. Please attach a covering letter giving details of the way in which you have handled each of the points raised by the referees. A revised manuscript will be once again subject to review and you probably understand that we can give you no guarantee at this stage that the eventual outcome will be favorable.

REFeree REPORTS

Reviewer #1:

The manuscript by Boyle et al. Describes the use of genome wide CRISPR screening across a large panel of cell lines to identify co-functional genes. This data has been extensively explored and described by several other groups in similar ways, identifying protein complexes, functional groups and correlated networks. The authors state that a major improvement of the current analyses is the result of "the power of unsupervised statistical techniques for correcting gene profiles....". However,

similar corrections have been described before and implemented in the analysis pipelines of the large screening data.

The manuscript by Boyle et al. is based on the analysis of Achilles CRISPR screens across 342 cell lines: "We first downloaded CRISPR screen gene summary data corrected for copy number confounding from the Project Achilles data depository (Meyers et al, 2017) The results of the CRISPR screens form a matrix, where each row in the matrix serves as a gene essentiality profile that summarizes the knockout phenotype of the gene across the 342 cell lines (the columns of the matrix)." At this moment in the Project Achilles data depository, the dataset Achilles CRISPR, vAvana-17Q2-Broad_v2 can be downloaded with the following description: "This dataset is the updated version of the Avana files published in Meyers, Bryan, et al. Computational correction of copy-number effect improves specificity of CRISPR-Cas9 essentiality screens in cancer cells. Nature Genetics. 2017. It has been changed as follows: - Cell line PK59_PANCREAS removed for quality control failure. " In addition, the README describes for this version the analysis steps, "10. Run CERES to infer gene scores and 11. Normalize gene scores with lists of core cell-essential and non-essential genes". These steps are not included in the changes from version V1 to V2 and the output of both versions are identical with respect to CERES gene effects. It is specifically stated that this data set is normalized to the positive and negative controls. It is unclear which dataset or version has been used for the analysis described in this manuscript but it seems that these data already corrected for the confounding effects of the differential sensitivity to CRISPR mediated double strand breaks and repair. If this is correct, one questions whether the authors want to hold on the hypothesis "However, found that the top four principal components explained significantly more variance than expected by permutation testing, with over 65% of the variance explained by the first principal component alone (Figure S1a). We hypothesize that the reason for these significant principal components is differential cell line sensitivity to dsDNA break toxicity". Because the remainder of the manuscript states that more accurate predictions of correlated gene essentiality profiles have been generated, this point has to be explained in more detail to support the final conclusion that "additional pre- and post-processing computational methods for genome-wide perturbation data is likely to advance progress towards a comprehensive understanding of cell network logic".

Reviewer #2:

The manuscript by Boyle et al. offers an analytical framework to derive co-functional gene networks from CRISPR-Cas9 screening data across cell lines (Project Achilles). They begin by assessing the effects of cell-line confounders in screening data, and after performing a PCA-based correction of these effects, they identify > 1 million pairs of genes with significantly correlated essentiality profiles. They proceed to demonstrate the utility of the network derived from these correlations by exploring several use cases, which include 1) drug-gene interactions, 2) network similarity with publicly-available datasets such as KEGG, CORUM, GO and STRING, 3) co-essential communities, and 4) network topology as it relates to disease annotations. The manuscript is well written and well referenced, including relevant preprints that were recently posted to bioRxiv, and the analyses are thorough and well executed.

The core innovation by the authors is their method of correcting cell-line-specific confounding effects, and its application to the Achilles dataset, to derive more accurate correlations between essentiality profiles. This is an important step, which to the extent of my knowledge, has not been implemented in other similar analyses. As such, the manuscript can be improved by providing a more systematic exploration of the possible biological or technical mechanisms underlying these artifacts, especially as the authors report that 65% of the corrected effect is concentrated in one Principal Component. The authors propose an explanation that differing sensitivity to DNA damage is the probable molecular mechanism. Can they provide additional evidence that this is the case? If the DNA damage hypothesis is true, perhaps TP53 status would be a potential biomarker that would explain part of it, or one could investigate the co-essential status of co-chromosomal genes as a marker for strong DNA damage effects.

While I do not think the results are going to differ much, the authors should consider rerunning their analyses on the latest Achilles dataset available, as it includes data for ~40% more cell lines. In fact, an analysis of the ability to identify co-functional genes as a function of the size of the dataset used

(i.e. down-sampling analysis) would provide an additional layer of validation to their methodology. Along these lines, an analysis of a few alternative correction methods would help make the case for the superiority of the proposed method. Specifically, I wonder how similar the correction would be when different sets of genes are used to perform PCA (curated non-essentials? Seldomly expressed? All genes?), and how well other common batch correction methods perform on this dataset.

The reason my suggestions center around a deeper exploration of the confounding effects is that the remaining analyses in the paper, those that focus on deriving biological function from the resulting correlation network, overlap with similar analyses that have been made public around the same time (particularly Kim et al, 2018, bioRxiv), which also evaluate significant correlations by deriving modular communities, evaluating functional enrichment and providing interesting biological connections to drug sensitivity. As such, deepening the focus on the main novelties in the analysis framework may raise the potential impact of this work.

It is of note that other works have not yet evaluated local centrality of genes and their relationship with disease. However, the authors mention the use of local centrality (as opposed to global) because of the potential overwhelming network degree of core growth genes. Does this effect not apply at the local level as well? It would be important to show that local centrality is not merely recapitulating mean KO essentiality within communities (the authors make a compelling case that this is already true at the global level in Figure 2E).

Minor comments:

The result in Figure S1d (the median cell line profile correlation dropping from ~0.85 to nearly 0 following correction) is counterintuitive for me. I'd expect the correction to keep, for example, core essential and non-essential genes as such, suggesting a significant correlation between essentiality profiles of different cell lines. Could it be that the corrected data version used here was gene-mean-centered?

Reference to Figure 6d in last Results section should be to Fig 7d.

Meyers et al.'s data depository is cited as ((Morgens et al, 2017)) in the first Methods paragraph.

I really like the schematics in Fig 1!

Reviewer #3:

Summary:

The authors present a PCA-based method for eliminating confounding artifacts in genetic perturbation screens, and bolstering sensitivity and specificity for detection of genetic interactions. The method was applied to a published set of >300 whole genome CRISPR screens and the authors report ~1 million pairs of correlated "co-functional" genes. A gene community approach was used to implicate core genes for cancer growth and compress signal from functionally related genes in the same community into a single score.

Overall, the manuscript provides an in-depth analysis of a large resource of essentiality screens that will serve to bolster human gene annotation through co-functionality.

Critique:

The manuscript is an important first step to control for technical confounding in large-scale CRISPR screening data across panels of genetically diverse cancer cell lines. The main assumption in the manuscript is that genes encoding olfactory receptors are expected to have little or no phenotype in the context of cancer cell line proliferation (i.e. good model of null phenotypes). Defining sets of essential and non-essential genes to guide the quality and performance of genome-wide loss-of-function screens has provided a framework for quantifying fitness effects in the past. It would have

useful to see the authors benchmark olfactory receptor genes with previously defined non-essential gene sets (e.g. Hart et al, MSB, 2014).

The authors begin by correcting for technical confounding found in parallel genetic screening data and observe that a large proportion of the variance is represented in the first 4 principal components, 68% of which is captured in PC1 (Fig S1a). The resulting PCs represent the likelihood that a given cell line is to exhibit essentiality amongst olfactory receptors. The authors should deconstruct PC1 and summarize the features that contribute to this massive effect. The text did not make it clear how exactly the confounding signatures were identified; that is, were PC1-PC4 used to calculate the confounding effect that was subtracted from the matrix of effect sizes? Or was it just PC1? Further clarification, deconstruction and summarization of the contributing factors to PC1 would have been highly informative and warrants some discussion in the results section.

The authors hypothesize that the reason for the significant PCs is differential cell line sensitivity to dsDNA break toxicity. I think this statement warrants some kind of analytical functional test. For example, are there correlates in PC1 features that suggest increased sensitivity to dsDNA break repair?

Construction of a set of co-functional genes from all pairs of human genes is an important goal for science. The authors do a good job at describing general and some specific observations of co-functionality. I couldn't get the Shiny app running for co-functional interaction visualization before submitting this review, so was not able to directly visualize the data. It would be helpful if the authors could provide a web portal for browsing/searching.

1st Revision - authors' response

26 October 2018

Reviewer #1:

The manuscript by Boyle et al. Describes the use of genome wide CRISPR screening across a large panel of cell lines to identify co-functional genes. This data has been extensively explored and described by several other groups in similar ways, identifying protein complexes, functional groups and correlated networks. The authors state that a major improvement of the current analyses is the result of "the power of unsupervised statistical techniques for correcting gene profiles....". However, similar corrections have been described before and implemented in the analysis pipelines of the large screening data.

We agree that the publication of the Project Achilles data we use is game-changing in the field, and dozens of other papers applying its data in some capacity have already been published; however, we believe this would be the first published example of a one-step preprocessing pipeline that dramatically improves co-essentiality estimation. Pan, et al (Cell Systems) adopts a carefully selected prefiltering step for ascertaining informative genes and applies arbitrary rank thresholds to detect protein complexes. Daley, et al (Genome Biology) very recently published a new likelihood approach for assessing hits that is much more technical but not immediately suited for co-essentiality studies.

The manuscript by Boyle et al. is based on the analysis of Achilles CRISPR screens across 342 cell lines: "We first downloaded CRISPR screen gene summary data corrected for copy number confounding from the Project Achilles data depository (Meyers et al, 2017) The results of the CRISPR screens form a matrix, where each row in the matrix serves as a gene essentiality profile that summarizes the knockout phenotype of the gene across the 342 cell lines (the columns of the matrix)." At this moment in the Project Achilles data depository, the dataset Achilles CRISPR, vAvana-17Q2-Broad_v2 can be downloaded with the following description: "This dataset is the updated version of the Avana files published in Meyers, Bryan, et al. Computational correction of copy-number effect improves specificity of CRISPR-Cas9 essentiality screens in cancer cells. Nature Genetics. 2017. It has been changed as follows: - Cell line PK59_PANCREAS removed for quality control failure. " In addition, the README describes for this version the analysis steps, "10. Run CERES to infer gene scores and 11. Normalize gene scores with lists of core cell-essential and non-essential genes". These steps are not included in the changes from version V1 to V2 and the output of both versions are identical with respect to CERES gene effects. It is specifically stated that

this data set is normalized to the positive and negative controls. It is unclear which dataset or version has been used for the analysis described in this manuscript but it seems that these data already corrected for the confounding effects of the differential sensitivity to CRISPR mediated double strand breaks and repair. If this is correct, one questions whether the authors want to hold on the hypothesis "However, found that the top four principal components explained significantly more variance than expected by permutation testing, with over 65% of the variance explained by the first principal component alone (Figure S1a). We hypothesize that the reason for these significant principal components is differential cell line sensitivity to dsDNA break toxicity". Because the remainder of the manuscript states that more accurate predictions of correlated gene essentiality profiles have been generated, this point has to be explained in more detail to support the final conclusion that "additional pre- and post-processing computational methods for genome-wide perturbation data is likely to advance progress towards a comprehensive understanding of cell network logic".

We agree completely with the reviewer. The draft submitted analyzed the original dataset from October 2017. It is reasonable to suspect that the updated processing steps improved the data quality and may have eliminated the advantage of our novel approach.

For this reason, we downloaded the newest release of Project Achilles (18Q2) and updated all correction and network figures. We find that the data quality has indeed improved: we now report over 1.5 million co-functional interactions with five rather than four signatures of confounding removed. The naïve approach also increased in yield, but the count increased from ~18,000 to ~31,000, illustrating that current measures adopted by Project Achilles are inadequate to attain the power we achieved.

We found that Figure 4e (clustering drug-gene associations by k-means) was unstable and replaced it with a simpler analysis where each drug's associations were analyzed separately. In Figure 7d PDGFRA no longer segregates in the NF1 community and was replaced with SOS1, a known RAS pathway gene. All other panels and results were essentially unchanged (e.g., Figure 4d now has three components instead of one connected component).

Figure 4e:

Figure 7d:

We wrote a new paragraph that replaces the analysis of the previous Figure 4e results:

For each drug compound, we tested the enrichment of KEGG terms amongst drug-gene associations. The most enriched term for each drug tended to confirm known biology (Figure 4e): Nutlin-3 upregulates p53 signaling, PLX4720 treats melanoma, and sorafenib modulates MAPK signaling. AZD6244 targets MAPK signaling pathways important for melanoma, explaining the extreme enrichment for ‘melanoma’ genes. Enrichment for SNARE proteins for Lapatinib associations suggests that RTK receptor trafficking is influencing the drug-gene network. Interestingly, L-685458, a gamma-secretase inhibitor, was most enriched for proteasome genes. There is debate in the literature whether gamma-secretase inhibitors, including L-685458, mediate their cancer killing effects via gamma-secretase or by the proteasome {Clementz:2009kb}. Cancer models often lack growth phenotypes when treated with gamma-secretase inhibitors, suggesting that these inhibitors hit multiple pathways such as the proteasome. In this panel it is thus plausible that proteasomal inhibition is the most important factor.

Reviewer #2:

The manuscript by Boyle et al. offers an analytical framework to derive co-functional gene networks from CRISPR-Cas9 screening data across cell lines (Project Achilles). They begin by assessing the effects of cell-line confounders in screening data, and after performing a PCA-based correction of these effects, they identify > 1 million pairs of genes with significantly correlated essentiality profiles. They proceed to demonstrate the utility of the network derived from these correlations by exploring several use cases, which include 1) drug-gene interactions, 2) network similarity with publicly-available datasets such as KEGG, CORUM, GO and STRING, 3) co-essential communities, and 4) network topology as it relates to disease annotations. The manuscript is well written and well referenced, including relevant preprints that were recently posted to bioRxiv, and the analyses are thorough and well executed.

We appreciate the reviewer’s kind feedback.

The core innovation by the authors is their method of correcting cell-line-specific confounding effects, and its application to the Achilles dataset, to derive more accurate correlations between essentiality profiles. This is an important step, which to the extent of my knowledge, has not been implemented in other similar analyses. As such, the manuscript can be improved by providing a more systematic exploration of the possible biological or technical mechanisms underlying these artifacts, especially as the authors report that 65% of the corrected effect is concentrated in one Principal Component. The authors propose an explanation that differing sensitivity to DNA damage is the probable molecular mechanism. Can they provide additional evidence that this is the case? If the DNA damage hypothesis is true, perhaps TP53 status would be a potential biomarker that would explain part of it, or one could investigate the co-essential status of co-chromosomal genes as a marker for strong DNA damage effects.

We have spent time examining the first principal component, as advised, and came to a somewhat unexpected conclusion. We first examined whether gene expression levels or mutational status significantly associated with the cell line loadings. We found that many genes’ expression level appeared to correlate with first PC loading; however, there was no functional enrichment among hits at a 5% FDR and the distribution was flat suggesting a mis-calibration of the null, such as low or correlated expression levels across genes, more than anything else. When we tested mutational status (both coding and annotated loss-of-function mutations), TP53 indeed exhibited the most significant association, but it still failed a 10% FDR cutoff and there did not seem to be signal for any related proteins amongst other top associated genes.

When we examined the variance of effect sizes in each cell line, the pattern was clear: 1st PC loadings were highly correlated with the variance, even more so when examining only olfactory receptors' effect sizes (now available as Figure EV1c).

A relatively recent preprint (McFarland, et al 2018, BioRxiv) suggests that rescaling and centering RNAi effect sizes can yield boosts in quality exceeding that associated with considering off target behavior. The high agreement between standard deviation of effect sizes and PC loading in each cell line suggests that this first PC might be re-centering and scaling the effect sizes, which should handle cases where a gene's effect sizes broadly have the same sign.

We have rewritten and extended this paragraph of the results section:

The loadings on the first principal component, which again explained most of the variance in olfactory receptor scores, were highly correlated with the variance in effect size estimates for olfactory receptor genes for each cell line ($R = -0.89$, Figure EV1c). One possibility is that regressing on the variance of each cell line acts as a technical correction, re-centering and scaling the effect sizes in a manner similar to that recommended in a recent preprint {McFarland:2018bv}. In this case, our scaling would roughly equalize variance in biological effect sizes amongst negative control genes. Alternatively, something that would be well captured by a single number per cell line such as cell health or sensitivity to double-strand breaks would also be consistent with prior work investigating CRISPR screen specificity and could produce the same relationship with variance in cell line effect sizes {Morgens:2017dw, Rosenbluh:2017hg}. In this case, one might expect mutational status in certain genes to predict the loading on cell line signatures, but the presence of coding or loss-of-function mutations in all recorded genes was not associated with principal component loadings (Wilcoxon rank-sum test, $q < 0.1$ for all genes). In any case, re-centering and scaling is not likely to underlie the four other orthogonal candidate signatures of confounding.

While I do not think the results are going to differ much, the authors should consider rerunning their analyses on the latest Achilles dataset available, as it includes data for ~40% more cell lines.

Our new manuscript features rerun analyses using the newest version of the Project Achilles CRISPR screen data (18Q2). See reviewer 1 comment above.

In fact, an analysis of the ability to identify co-functional genes as a function of the size of the dataset used (i.e. down-sampling analysis) would provide an additional layer of validation to their methodology.

Yes, downsampling can be informative for understanding how many screens are necessary to apply this approach. Interestingly, around 100 screens are needed to achieve robust performance, and it steeply increases as more screens are added. At 436, the number of interactions appears to begin to plateau, but if more heterogeneous screens or cell types are added, the interactions called may change. Figure EV1g illustrates the analysis and is now referenced in the text.

Across all 17,634 genes tested, 1,528,726 co-functional interactions, equal to 0.98% of all possible pairs of genes, met a false discovery rate of 10%. A similar procedure on raw gene essentiality profiles yielded only 30,761 interactions. Downsampling of the number of cell lines included in the analysis shows that more cell lines tighten the null distribution and increase power to discover co-functional interactions (Figure EV1g).

Along these lines, an analysis of a few alternative correction methods would help make the case for the superiority of the proposed method. Specifically, I wonder how similar the correction would be when different sets of genes are used to perform PCA (curated non-essentials? Seldomly expressed? All genes?), and how well other common batch correction methods perform on this dataset. The reason my suggestions center around a deeper exploration of the confounding effects is that the remaining analyses in the paper, those that focus on deriving biological function from the resulting correlation network, overlap with similar analyses that have been made public around the same time (particularly Kim et al, 2018, bioRxiv), which also evaluate significant correlations by deriving modular communities, evaluating functional enrichment and providing interesting biological connections to drug sensitivity. As such, deepening the focus on the main novelties in the analysis framework may raise the potential impact of this work.

The sensitivity of the analysis to the negative control gene set used is a key consideration. We repeated the co-functional interaction calling with both Hart-curated nonessential genes and all genes with an average FPKM expression level under 0.1. Both cases yielded very similar loadings on the first principal component and a comparable number of co-functional interactions (the number of significant principal components increased from 5 to 6). The figure comparing loadings from olfactory receptors vs. nonessential genes is now available as Figure EV1b:

We have added a sentence to the text including this information:

However, we found that the top five principal components explained significantly more variance than expected by permutation testing, with over half of the variance explained by the first principal component alone (Figure EV1a). We repeated this approach with a curated set of nonessential genes in place of olfactory receptors and reproduced the loadings on the first principal component, demonstrating that signatures are robust to choice of control gene set ($R = 0.92$, Figure EV1b).

It is of note that other works have not yet evaluated local centrality of genes and their relationship with disease. However, the authors mention the use of local centrality (as opposed to global)

because of the potential overwhelming network degree of core growth genes. Does this effect not apply at the local level as well? It would be important to show that local centrality is not merely recapitulating mean KO essentiality within communities (the authors make a compelling case that this is already true at the global level in Figure 2E).

It is true that there is a relationship between local closeness and essentiality. The mean growth phenotype per community and the local closeness are correlated with $R^2 = 0.246$ genome-wide. However, global network degree is correlated with mean growth phenotype per community with $R^2 = 0.731$. Thus, local closeness mitigates the powerful influence of very essential genes when measured genome-wide. Future network-based analyses may shed light on how best to conceive of gene modules in the context of CRISPR screens.

Minor comments:

The result in Figure S1d (the median cell line profile correlation dropping from ~0.85 to nearly 0 following correction) is counterintuitive for me. I'd expect the correction to keep, for example, core essential and non-essential genes as such, suggesting a significant correlation between essentiality profiles of different cell lines. Could it be that the corrected data version used here was gene-mean-centered?

Reference to Figure 6d in last Results section should be to Fig 7d.

“Figure 6d” has been changed to “Figure 7d”

Meyers et al.'s data depository is cited as ((Morgens et al, 2017)) in the first Methods paragraph.

The line now cites Meyers et al instead of Morgens et al.

We first downloaded CRISPR screen gene summary data corrected for copy number confounding from the Project Achilles data depository (Meyers *et al*, 2017) and matched RNA-seq and mutation data from the Cancer Cell Line Encyclopedia (CCLE) website (Barretina *et al*, 2012).

I really like the schematics in Fig 1!

Thanks!

Reviewer #3:

Summary:

The authors present a PCA-based method for eliminating confounding artifacts in genetic perturbation screens, and bolstering sensitivity and specificity for detection of genetic interactions. The method was applied to a published set of >300 whole genome CRISPR screens and the authors report ~1 million pairs of correlated "co-functional" genes. A gene community approach was used to implicate core genes for cancer growth and compress signal from functionally related genes in the same community into a single score.

Overall, the manuscript provides an in-depth analysis of a large resource of essentiality screens that will serve to bolster human gene annotation through co-functionality.

We are glad to hear the reviewer appreciates our contribution to new approaches gene co-functionality.

Critique:

The manuscript is an important first step to control for technical confounding in large-scale CRISPR screening data across panels of genetically diverse cancer cell lines. The main assumption in the manuscript is that genes encoding olfactory receptors are expected to have little or no phenotype in

the context of cancer cell line proliferation (i.e. good model of null phenotypes). Defining sets of essential and non-essential genes to guide the quality and performance of genome-wide loss-of-function screens has provided a framework for quantifying fitness effects in the past. It would have been useful to see the authors benchmark olfactory receptor genes with previously defined non-essential gene sets (e.g. Hart et al, MSB, 2014).

We have now directly compared the signature of confounding learned from olfactory receptors to that learned using the aforementioned set of non-essential genes, seen in Figure EV1b, and report little difference. See also response to reviewer #2 above.

The authors begin by correcting for technical confounding found in parallel genetic screening data and observe that a large proportion of the variance is represented in the first 4 principal components, 68% of which is captured in PC1 (Fig S1a). The resulting PCs represent the likelihood that a given cell line is to exhibit essentiality amongst olfactory receptors. The authors should deconstruct PC1 and summarize the features that contribute to this massive effect. The text did not make it clear how exactly the confounding signatures were identified; that is, were PC1-PC4 used to calculate the confounding effect that was subtracted from the matrix of effect sizes? Or was it just PC1? Further clarification, deconstruction and summarization of the contributing factors to PC1 would have been highly informative and warrants some discussion in the results section.

The authors hypothesize that the reason for the significant PCs is differential cell line sensitivity to dsDNA break toxicity. I think this statement warrants some kind of analytical functional test. For example, are there correlates in PC1 features that suggest increased sensitivity to dsDNA break repair?

To address the reviewer's concern, we have performed more rigorous analysis of what underlies the first principal component's signature, which is the most impactful. See the response to reviewer #2 above and the new panels in Figure EV1. We have also clarified that all four principal components (with the larger dataset now five are significant) are retained and subtracted from the matrix:

To produce an improved dataset, we first subtracted all five candidate signatures of confounding from each gene's essentiality profile (Figure 1c). Using these corrected gene essentiality profiles, we again computed the correlation of all pairs of genes.

Construction of a set of co-functional genes from all pairs of human genes is an important goal for science. The authors do a good job at describing general and some specific observations of co-functionality. I couldn't get the Shiny app running for co-functional interaction visualization before submitting this review, so was not able to directly visualize the data. It would be helpful if the authors could provide a web portal for browsing/searching.

A web portal is now available at https://greenleaf.shinyapps.io/cofunctional_app_20181015/. Happy browsing!

2nd Editorial Decision

22 November 2018

Thank you again for sending us your revised manuscript. We have now heard back from reviewer #2 who agreed to evaluate your study. As you will see below, the reviewer thinks that all issues have been satisfactorily addressed and is supportive of publication.

Before we formally accept the study for publication, we would ask you to address minor editorial issues.

REFeree REPORTS

Reviewer #2:

The authors fully addressed my previous concerns and I have no additional ones. Very nice work!

Corresponding Author Name: William Greenleaf

Manuscript Number: MSB-18-8594